# Innovative Microbial Immobilization Strategy for Di-*n*-Butyl Phthalate Biodegradation Using Biochar-Calcium Alginate-Waterborne Polyurethane Composites

**DOI:** 10.3390/microorganisms12071265

**Published:** 2024-06-22

**Authors:** Xuan-Di Cao, Shih-Hao Jien, Chu-Wen Yang, Yi-Hsuan Lin, Chien-Sen Liao

**Affiliations:** 1Institute of Biotechnology and Chemical Engineering, I-Shou University, Kaohsiung 840203, Taiwan; isu10937051d@cloud.isu.edu.tw; 2Department of Soil and Environmental Sciences, National Chung Hsing University, Taichung 402202, Taiwan; shjien@nchu.edu.tw; 3Department of Microbiology, Soochow University, Taipei 111002, Taiwan; ycw6861@scu.edu.tw; 4Environmental Engineering Research Center, Sinotech Engineering Consultants Inc., Taipei 114065, Taiwan; yhlin@sinotech.org.tw; 5Department of Medical Science and Biotechnology, I-Shou University, Kaohsiung 824005, Taiwan; 6Institute of Biopharmaceutical Sciences, National Sun Yat-sen University, Kaohsiung 804201, Taiwan

**Keywords:** biochar, calcium alginate, waterborne polyurethane, microbial immobilization, *Bacillus aquimaris*, di-*n*-butyl phthalate

## Abstract

Di-*n*-butyl phthalate (DBP) is a prevalent phthalate ester widely used as a plasticizer, leading to its widespread presence in various environmental matrices. This study presents an innovative microbial immobilization strategy utilizing biochar, calcium alginate (alginate-Ca, (C_12_H_14_CaO_12_)n), and waterborne polyurethane (WPU) composites to enhance the biodegradation efficiency of DBP. The results revealed that rice husk biochar, pyrolyzed at 300 °C, exhibits relatively safer and more stable physical and chemical properties, making it an effective immobilization matrix. Additionally, the optimal cultural conditions for *Bacillus aquimaris* in DBP biodegradation were identified as incubation at 30 °C and pH 7, with the supplementation of 0.15 g of yeast extract, 0.0625 g of glucose, and 1 CMC of Triton X-100. Algal biotoxicity results indicated a significant decrease in biotoxicity, as evidenced by an increase in chlorophyll *a* content in *Chlorella vulgaris* following DBP removal from the culture medium. Finally, microbial community analysis demonstrated that encapsulating *B. aquimaris* within alginate-Ca and WPU layers not only enhanced DBP degradation, but also prevented ecological competition from indigenous microorganisms. This novel approach showcases the potential of agricultural waste utilization and microbial immobilization techniques for the remediation of DBP-contaminated environments.

## 1. Introduction

Phthalate esters (PAEs) play a crucial role as industrial chemicals widely utilized as plasticizers in various applications, such as food plastic packaging, cosmetics, furniture decoration materials, chemical fertilizers, and the pharmaceutical industry [1,2,3,4,5]. Operating as plasticizers, PAEs significantly improve the mechanical properties of plastic resin, particularly enhancing flexibility. However, PAEs are not chemically bound to the plastic matrix [6,7], making them susceptible to detachment from plastic products and dispersion into the surrounding environment. A 2023 study revealed seasonal fluctuations in PAE concentrations within ponds across diverse regions in India, attributing these variations to seasonal changes in temperature and precipitation [8]. Moreover, research identified substantial PAEs pollution in the waters of the Persian Gulf in Iran, significantly impacting the ecological environment of the coral reefs in the area [9]. Similarly, three common PAEs were detected in the waters of Poyang Lake, China, with their concentrations posing ecological risks [10]. In 2019, a South Korean research team discovered the presence of four common plasticizers in seawater along the southeastern coastal areas, attributing their origins to anthropogenic and industrial emissions [11]. During the same period, various plasticizers were detected in river sediments and fish bodies in Taiwan [12], with concentrations exhibiting minimal divergence from those recorded at the same locations eighteen years earlier [13]. Recent years have highlighted the endocrine-disrupting effects of PAEs, associating them with adverse health outcomes in both animals and humans. Consequently, the United Nations (UN) and governmental agencies in several countries have classified six PAEs as priority environmental pollutants and endocrine-disrupting compounds (EDCs).

DBP stands out as one of the most frequently encountered PAEs, primarily being incorporated into cellulose-based plastics, serving as a coalescing aid in latex adhesives, and functioning as a solvent for dyes [14]. Previous studies have identified the presence of DBP in various sources, including drinking water [15], food [16], agricultural fields [17], and even in pharmaceuticals [18]. The environmental release of DBP can lead to its uptake by crops, subsequently entering the food chain. Numerous studies examining the biotoxicity of DBP have been conducted. Exposure to DBP in barley seedlings resulted in significant alterations in biochemical indices, particularly in the roots [19]. When combined with polystyrene, DBP further reduced lettuce biomass and induced oxidative stress [20]. In rats, DBP exposure induced oxidative stress, DNA oxidation, and neuroinflammation, culminating in neuronal damage [21]. Moreover, DBP was found to inhibit growth and induce significant changes in biosynthesis-relevant proteins in *Chlorella vulgaris*, a type of microalgae [22]. Collectively, these studies underscore the biotoxicity of DBP across different species, emphasizing its potential to inflict significant harm on both the environment and human health.

A range of studies have explored the bioremediation of DBP in different contexts. Baker and Feng both isolated bacteria, *Rhodovulum* sp. DBP07 and *Pseudomonas* sp. YJB6, respectively, that demonstrated high DBP-degrading activity [23,24]. These bacteria were found to degrade DBP into intermediate compounds, with *Pseudomonas* sp. YJB6 showing a complete biodegradation pathway. Kong further demonstrated the bioremediation potential of *Gordonia* sp. strain QH-11 in agricultural soils, which not only accelerated DBP degradation, but also influenced the native microbial community and ecosystem functions [25]. Mphahlele demonstrated the potential of a dual biological and photocatalytic system for DBP removal in wastewater, achieving a 69% degradation [26]. Liu investigated the removal of DBP from aged landfill leachate, achieving a 96% removal rate with the optimal hydraulic conditions and the presence of dominant DBP-degrading bacteria [27]. Ziembowicz explored the use of Fenton-like processes, catalyzed by natural and modified bottom sediments, for DBP degradation in landfill leachate, achieving almost 95% degradation [28]. These studies collectively highlight the potential of various bioremediation approaches for DBP, including biological, photocatalytic, and Fenton-like processes.

Microbial immobilization technology (MIT) is a methodology that employs chemical or physical approaches to restrict or localize free cells or enzymes within a specific spatial region while preserving their biological activity. Due to its remarkable efficiency and stability, MIT has garnered considerable attention for the treatment of wastewater [29]. Studies have explored the use of MIT for enhancing the biodegradation of phthalates. Bera demonstrated the potential of natural matrices like areca nut husks and luffa sponge for immobilizing a bacterial consortium capable of degrading phenol [30]. Wang and Zhang employed immobilization methods using magnetic nanoparticles and corncob-sodium alginate, respectively, to enhance the biodegradation of DBP and di-*n*-octyl phthalate by *Acinetobacter* species and *Burkholderia* sp. strain ETG-101, respectively [31,32]. Additionally, Wang isolated *Achromobacter* sp. RX that was capable of degrading di-(2-ethylhexyl) phthalate and demonstrated its potential for decontaminating DEHP-contaminated soil [33]. In recent years, the application of biochar-immobilized microorganisms for pollutant removal from wastewater has gained significant attention. Biochar, known for its high porosity and strong adsorption capacity, serves as an efficient carrier for microbial immobilization. Various methods, including adsorption, embedding, and electrochemical immobilization, have been explored to enhance the removal efficiency of pollutants. The interaction mechanisms between biochar and microorganisms, particularly focusing on electrostatic interaction and ion grid formation, play a crucial role in the immobilization process [34]. These studies collectively underscore the potential of microbial immobilization, including the use of biochar, for enhancing phthalate biodegradation. However, despite the protective effect of the immobilized carrier on microorganisms, no reports to date have addressed potential modifications in bacteria within immobilized particles during operation [29].

The objective of this study is to design a novel microbial immobilization method for DBP bioremediation utilizing biochar, alginate-Ca, and WPU. Rice husk was subjected to high-temperature pyrolysis to produce biochar. The physical and chemical characteristics of the biochar were analyzed to determine the optimal preparation conditions. Subsequently, the biochar, adsorbed with DBP-degrading bacteria (*B. aquimaris*), was encapsulated using alginate-Ca and WPU, forming immobilized particles. The physical and chemical properties of these immobilized particles were then analyzed to identify the optimal preparation conditions. Finally, optimal cultivation conditions for DBP biodegradation were analyzed and established. Algal biotoxicity analysis of degradation metabolites were conducted using *C. vulgaris*. Next-Generation Sequencing (NGS) technology was employed to assess the variations in the microbial community within the immobilized particles during the DBP degradation process. The outcome of this study is to provide an application model for the rapid and effective removal of DBP.

## 2. Materials and Methods

### 2.1. Chemicals

The standard reagent, DBP (98.7% purity, CAS: 84-74-2), was purchased from Riedel-deHaën Co., Berlin, Germany. The solvents utilized included acetone and n-hexane, both purchased from E. Merck, Darmstadt, Germany. Sodium alginate was acquired from Tong Ho Chemical & Enterprise Co., Taipei, Taiwan, and WPU was procured from Gabriel Advanced Materials Co., Taipei, Taiwan. Trypticase soy agar, trypticase soy broth, and R_2_A agar were purchased from Becton, Dickinson, and Co., Franklin Lakes, NJ, USA. Chlorophyll *a* standard (from spinach, CAS: 479618), and other additional chemicals were all purchased from E. Merck, Darmstadt, Germany. All reagents were of reagent HPLC-grade quality. DBP stock solutions were prepared by dissolving DBP in acetone to a concentration of 10,000 mg L^−1^. Glassware was meticulously cleaned to minimize PAE contamination, washed with deionized water, dried overnight at 80 °C, rinsed twice with acetone, and air-dried before use.

### 2.2. Isolation and Identification of DBP-Degrading Bacteria

River water and sediment samples were collected from the Hou-Jing River (22°00′42.031″ N, 120°00′19.033″ E), located in Kaohsiung, Taiwan, an area characterized by several industrial zones and plastic industries upstream of the Hou-Jing River. Both the river water and sediment exhibited slight contamination from plasticizers. All water and sediment samples were collected in 3 L sterile glass bottles and were stored at 4 °C. The samples were manually mixed until visually homogeneous, and then were selected for analysis. Bacteria from river water and sediment samples, exhibiting the capability to degrade DBP, were cultured on Trypticase soy agar and R2A agar. Following 48 h of incubation at 30 °C, distinct colonies of varying morphological types emerged, which were subsequently streaked onto fresh TSA plates containing 5 mg L^−1^ of DBP to isolate the degrading bacteria. Thereafter, the DBP-degrading bacteria were subjected to 16S rRNA identification. The primers utilized to amplify the 16S rRNA gene were Pf: 5′-AGAGTTTGATCCTGGCTCAG-3′ and Pr: 5′-ACGGCTACCTTGTTACGACT-3′, corresponding to positions 8–27 and 1495–1514, respectively, of the coli 16S rRNA gene. PCR was employed for the amplification process, and the resultant products were subjected to gel electrophoresis in 1% agarose. DNA sequencing services were conducted by Seeing Bioscience Co., Ltd. (Taipei, Taiwan).

### 2.3. Preparation and Characterization of Biochar

#### 2.3.1. Biochar Production

The biochar utilized in this study was derived from rice husks (RH), a significant agricultural byproduct generating 0.3 million tons annually in Taiwan. The RH were sourced from the Kaohsiung District Agricultural Research and Extension Station, Ministry of Agriculture of Taiwan. Prior to carbonization, the rice husks were dried at 60 °C for 24 h until reaching a moisture content of less than 10%. For the pyrolysis process, the samples were introduced into a tubular furnace fitted with a corundum tube (32 mm in diameter, 700 mm in length) and subjected to an N2 purge (1 L min^−1^ flow rate) to maintain an oxygen-free environment. Pyrolysis was carried out at temperatures of 300 °C, 450 °C, and 600 °C, respectively, with a heating rate of 5 °C min^−1^. The temperature was sustained for 2 h before gradually cooling to ambient temperature under N2 flow. The resulting rice husk biochar (RHB) was designated as RHB-300, RHB-450, and RHB-600 in subsequent sections of this study. Following pyrolysis, the RHB was finely ground to pass through a 2 mm sieve, ensuring uniform particle size across all samples used in the experiments.

#### 2.3.2. Basic Properties of the RH and RHB

The pH of the RH and RHB were determined in a mixture with deionized water (1:10 *w*/*v*) using a glass electrode [35]. Cation exchange capacity (CEC) was determined using the ammonium acetate method (pH 7.0) [36]. Total carbon and nitrogen contents were measured with a Fisons NA1500 elemental analyzer (Thermo Electron Corporation, Waltham, MA, USA). Water content was determined with the indirect drying method [37]. A scanning electron microscope (SEM, Philips 501 scanning electron micrograph; Philips, Eindhoven, The Netherlands) was used to characterize the morphology of the RH and RHB.

#### 2.3.3. Chemical Compositions Analysis

The chemical composition of RH and RHB surfaces is determined by analyzing the characteristic X-ray spectrum emitted by the specimen under examination [38]. Chemical composition analysis was conducted in a “spot mode,” focusing the beam on a manually selected single area within the field of view. The specific location is indicated on the accompanying SEM images with a cross symbol. The energy dispersive spectroscopy (EDS) detector used in this study was capable of detecting elements with atomic numbers equal to or greater than six. It is important to note that the intensity of peaks in the EDS spectrum does not directly correlate with elemental concentration; however, relative amounts can be inferred from peak heights. By examining the peaks in the EDS spectrum and comparing them to known elemental signatures, the elements present in the samples surface can be identified.

### 2.4. Immobilization Methods

After cultivating the test bacterial suspension in trypticase soy broth, the bacterial density was allowed to reach O.D.600 = 0.6. Subsequently, the RHB-300, RHB-450, and RHB-600 were immersed in bacterial suspension for 48 h to enable adsorption onto biochar. Different ratios (0.5, 2, 3%) of alginate-Ca solutions (100 mL) were prepared and mixed with the RHB that had adsorbed the bacterial suspension for 6 h. Additionally, a 5% calcium chloride solution was prepared and placed on a digital stirrer. The mixture containing biochar and 2% sodium alginate solution was then dropwise added into the calcium chloride solution, forming spherical alginate-Ca immobilized particles. After one hour, the calcium chloride solution was filtered, washed with RO water, and the external surface of the alginate-Ca microbial immobilization particles was coated with a 60% concentration of WPU, resulting in the formation of immobilized particles known as alginate-Ca/WPU.

### 2.5. Analysis of Microbial Immobilized Particles

The water content, weight, and weight loss ratio of individual microbial immobilized particles produced using different ratios of alginate-Ca/WPU were assessed. Water content was determined using the indirect drying method. Ten microbial immobilized particles were subjected to drying in a laboratory oven with forced air circulation and thermostatic temperature control at 110 °C for 90 min. Following drying, the samples were transferred to a desiccator, cooled, and then weighed using an analytical balance. The initial weight of the particles was subtracted from the weight obtained after drying. The difference was divided by the initial weight, multiplied by 100%, and divided by ten to determine the water content of a single microbial immobilized particle after drying.

The dried microbial immobilized particles mentioned above were placed into 125 mL serum bottles at a weight of 5 g each. Subsequently, 20 mL of microbial culture medium (the composition of which will be detailed in the next section) was added, and the bottles were then placed in a constant temperature incubator for cultivation. Cultivation was conducted at 28 °C with shaking at 150 rpm for 48 h. Afterward, the particles were removed and dried to remove water. Finally, the weight loss ratio was calculated by subtracting the weight of the particles after 48 h of shaking from the weight of the particles measured on the first day, dividing by the weight of the particles on the first day, and multiplying by 100%.

### 2.6. Experimental Design

Enrichment cultures of pure DBP-degrading bacteria (free cells) were employed in conjunction with Trypticase soy broth. The microbial culture medium comprised the following components (all concentrations in mg L^−1^): K_2_HPO_4_, 65.3; KH_2_PO_4_, 25.5; Na_2_HPO_4_·12H_2_O, 133.8; NH_4_Cl, 5.1; CaCl_2_, 82.5; MgSO_4_·7H_2_O, 67.5; and FeCl_3_·6H_2_O, 0.75. The pH of the medium was adjusted to 7 using potassium hydroxide before autoclaving at 121 °C for 25 min. Batch experiments were conducted in 125 mL glass bottles containing 45 mL of microbial culture medium and either 5 mL of pure DBP-degrading bacteria culture medium or 5 g of immobilized particles. Each experimental bottle was supplemented with 5 mg L^−1^ of DBP. The following incubation factors were then altered to investigate their impact on DBP biodegradation in the medium: cultural temperature (25, 30, 35, and 40 °C), initial pH levels (5, 6, 7, 8, and 9), carbon and nitrogen sources (yeast extract 0.25 g, yeast extract 0.15 g + glucose 0.125 g, yeast extract 0.15 g + glucose 0.0625 g, or yeast extract 0.15 g + glucose 0.03125 g), surfactants (Brij 35, Triton X-100, Tergitol, Triton N101, and Triton X-80), and shaking rate (static or 160 rpm). The surfactants were all prepared at 1 critical micelle concentration (CMC). After determining optimal conditions, DBP biodegradation under these conditions in the culture medium and original wastewater was also analyzed. Incubation occurred at 30 °C in the dark under aerobic conditions, and aqueous samples were periodically collected to measure residual DBP concentrations.

### 2.7. Analysis of Residual DBP

Residual DBP in the culture medium was analyzed according to the method of Liao [22], with slight modifications. In brief, 2 mL of culture medium was combined with 2 mL of n-hexane in bottles and shaken at 160 rpm for 2 h. The extraction of residual DBP was conducted in three repetitions, and the resulting extracts were combined for analysis. The analysis was performed using a Perkin Elmer Clarus 400 gas chromatograph coupled with an electron capture detector (PerkinElmer, Inc., Waltham, MA, USA) and an Elite-5 ms capillary column (film thickness, 0.25 μm; inner diameter, 0.25 mm; length, 30 m). The injector temperature was maintained at 250 °C. Nitrogen served as the carrier gas with a flow rate of 0.8 mL min^−1^ at a 10:1 split ratio. The initial column temperature was set at 150 °C for 1 min, followed by an increase at a rate of 6 °C min^−1^ to 220 °C, then a further increase at a rate of 3 °C min^−1^ to 275 °C, where it was held for 13 min. The detector temperature was set at 320 °C. The recovery percentage of DBP was 96.5%, and the method detection limit (MDL) was 90 μg L^−1^.

### 2.8. Algal Biotoxicity Assessments

Algal biotoxicity assessments utilized the freshwater unicellular green alga *C. vulgaris* Beij. #3001 due to its high sensitivity to contaminants. The algae were cultured in the medium developed by Liao [39], maintaining all cultures at 25 °C with an illumination of approximately 400 μmol·m^−2^·s^−1^ and a 14:10 h light-dark cycle. Cultures were consistently diluted every two days to sustain log-phase growth, maintaining a density of 5–20 micrograms of chlorophyll a per liter. Chlorophyll *a* content in *C. vulgaris* in the microbial culture medium was quantified spectrophotometrically at absorbances of 680 and 750 nm before and after DBP biodegradation experiments, respectively. Median effective concentration (EC_50_) values, indicative of DBP biotoxicity to *C. vulgaris*, were determined by evaluating the inhibition of cell growth. Toxicity calculations were based on the reduction in chlorophyll *a* content after the 24 h test period. A blank test without DBP was conducted as a control, and toxicity in this scenario was analyzed after the 24 h test period. Statistical calculations were based on a minimum of three repeated tests, and each test was performed with three replicate cultures.

### 2.9. Microbial Community Analysis

This study utilizes NGS technology to analyze environmental microbial communities. DBP-degrading bacteria, isolated from river samples, were cultured under optimal conditions for DBP biodegradation. After 48 h of incubation, 5 mL of culture medium or 5 g of immobilized particles were added to 45 mL of original river water supplemented with 5 mg L^−1^ DBP. DBP was not detectable or was below the detection limit in the original river water. The mixture was then incubated for 7 days to evaluate the survival of DBP-degrading bacteria in the natural environment. DNA extractions from the samples were carried out using the PowerWater DNA Isolation kit (QIAGEN, Venlo, The Netherlands). The V5-V8 variable regions of the 16S rRNA gene were amplified using a 5′ primer with the 16S rRNA gene-specific sequence 341F (5′-CCTACGGGNBGCASCAG-3′) and sequencing adaptor (5′-TCGTCGGCAGCGTCAGATGTGTATAAGAGACAG-3′). The 3′ primer comprised the sequencing adaptor (5′-GTCTCGTGGGCTCGGAGATGTGTATAAGAGACAG-3′) and the 16S rRNA gene-specific sequence 805R (5′-GACTACNVGGGTATCTAATCC-3′). PCR reactions, with a 25 μL PCR mixture, included PCR buffer, 200 mM of each deoxynucleotide triphosphate, 10 pmol of each primer, 1.25 U of Taq polymerase, and 50 ng of template DNA. PCR conditions were as follows: 95 °C for 10 min, 30 cycles of 95 °C for 1 min, 55 °C for 1 min, 72 °C for 1 min, and a final step at 72 °C for 15 min. PCR products were verified using 1.2% (*w*/*v*) agarose gel electrophoresis. Amplicon sequencing of the 16S rRNA was conducted on the MiSeq platform (Illumina, Inc., San Diego, CA, USA). NGS analysis was performed at the Seeing Bioscience Co., Ltd. (Taipei, Taiwan), and data analysis followed previously described methods [40].

### 2.10. Statistical Analyses

The DBP degradation data collected for this study fit well with first-order kinetics: S = S_0_exp(−*k*_1_*t*), t_1/2_ = ln 2*k*_1_^−1^, where S_0_ is the initial concentration, S is the substrate concentration, *t* is the time period, and *k*1 is the degradation rate constant. The remaining percentage was calculated as the DBP residue concentration divided by the original DBP concentration, multiplied by 100. Each treatment was performed in triplicate. Statistical analysis was carried out using ANOVA.

## 3. Results

### 3.1. Identification of DBP-Degrading Strain

Twenty-five bacterial strains capable of utilizing DBP as the carbon source and energy were isolated from water and sediment samples collected from the Hou-Jing River. Incubation was conducted at 30 °C, pH 7.5, with an initial DBP concentration of 5 mg L^−1^. After 48 h of incubation, three isolates, designated as strains HJ-6, HJ-14, and HJ-21, exhibited significant DBP degradation capability, whereas the remaining twenty-two strains showed negligible DBP degradation capability. Following identification using 16S rRNA sequencing, these three isolates were classified as *B. aquimaris*, *Rhodococcus ruber*, and *Hydrogenophaga bisanensis*, respectively. Subsequently, after comparing their DBP degradation abilities, *B. aquimaris* (HJ-6), exhibiting the strongest degradation capability, was selected as the experimental strain for this study.

### 3.2. Basic Properties of Biochar

The basic property analysis of biochar, as shown in Table 1, revealed notable differences between RH and RHB (RHB-300, RHB-450, and RHB-600). The pH of RH shifted from weakly acidic to weakly alkaline after pyrolysis. Parameters such as Total Carbon, Total Nitrogen, Water Content, and CEC decreased, while the C/N ratio showed a slight increase. This phenomenon suggested that rice hull transformed into rice hull biochar with a higher C/N ratio post-pyrolysis. Based on our previous studies [41,42], they displayed SEM observations of rice husk biochar (RHB) and wood biochar (WB), which were pyrolyzed from different temperatures. They indicated that more pores with diameter ≤ 10 μm were found in the biochars with a higher pyrolyzation temperature (600 °C) than the lower one (400 °C). Pyrolysis processing of biomass enlarges the crystallites and makes them more ordered, and this effect increases with HTT. Lua et al. indicated that increasing the pyrolysis temperature from 250 °C to 500 °C increases the BET surface area due to the greater evolution of volatiles from pistachio-nut shells, resulting in enhanced pore development in biochar [43].

In the chemical composition analysis of RH and RHB, the results from Table 2 indicate that, among the biochars produced at different pyrolysis temperatures, RHB-300 exhibited the highest proportion of oxygen in its surface chemical composition. As the pyrolysis temperature increased, the proportion of oxygen gradually decreased and specific surface area (SSA) increased. The increase in SSA on the biochar may lead to an increase in the adsorption of chemicals such as PAHs. To avoid concerns regarding excessive compound presence on the biochar surface during the fabrication of microbial immobilized particles, our study opted for RHB-300, which might be a less toxic substance on the surface compared with RHB-600, as the raw material for preparing microbial immobilized particles.

The appearance of the RH, RHB, alginate-Ca, and alginate-Ca/WPU immobilized particles are presented in Figure 1. In Figure 1D, it is evident that Alginate-Ca/WPU immobilized particles, with WPU coating, display a thin white film layer compared to Alginate-Ca particles. SEM images of RH and RHB are shown in Figure 2. These images revealed porous surfaces for both RH and RHB. SEM images of RHB-300, immersed in a bacterial suspension for 48 h, are shown in Figure 3. The white arrows in the images indicate that, after immersion in the bacterial suspension, the surface and pores of RHB-300 became covered with numerous bacteria.

### 3.3. Characterization of Microbial Immobilized Particles

Table 3 shows the water content and weight of a single alginate-Ca/WPU microbial immobilized particle which was prepared with different ratios of alginate-Ca. The analysis of the single particle’s weight showed that particles at 0.5% alginate-Ca concentration weighed 31.31 ± 0.85 mg, those at 2% weighed 55.94 ± 1.34 mg, and those at 3% weighed 68.65 ± 1.88 mg. Regarding the water content analysis, particles at 0.5% alginate-Ca concentration had a water content of 18.96%, those at 2% had 26.45%, and those at 3% had 35.66%. These results indicate that lower concentrations of alginate-Ca correspond to lower water content and particle weight.

Table 4 shows the results of the weight loss test for microbial immobilized particles. Following 48 h of shaking, the weight loss ratio was 12.23% for 0.5% alginate-Ca particles, 10.52% for 2% alginate-Ca particles, and 8.23% for 3% alginate-Ca particles. After 96 h of shaking, the weight loss ratio was 23.17% for 0.5% alginate-Ca-concentrated particles, 18.42% for 2% alginate-Ca particles, and 15.21% for 3% alginate-Ca particles. These experimental findings indicated that higher alginate-Ca concentrations correspond to lower weight loss ratios. The analysis results from Table 3 and Table 4 revealed that, in alginate-Ca/WPU microbial immobilized particles, lower alginate-Ca concentrations correlate with lower water content and particle weight, while higher alginate-Ca concentrations correspond to lower weight loss ratios. Preliminary testing indicated that alginate-Ca concentrations below 0.5% resulted in excessively soft particles with poor sphericity, whereas concentrations above 3% led to overly rigid particles, diminishing their effectiveness in biodegradation applications. Consequently, this study selected 0.5%, 2%, and 3% concentrations for testing. Given the goal of achieving higher water content and lower weight loss ratios in microbial immobilized particles for optimal microbial environments, this study ultimately chose a 3% alginate-Ca concentration for subsequent particle preparation.

### 3.4. Optimal Cultural Conditions of DBP Biodegradation

Table 5 shows the effects of various incubation factors on the biodegradation rate constants (*k*_1_, day^−1^) and half-lives (t_1/2_, days) of DBP by free cells and two types of immobilized particles. During the DBP biodegradation experiments, we observed that the microbial degradation of DBP fit well with first-order kinetics. Based on the characteristics of first-order kinetics, we utilized the statistical formulas provided in Section 2.10 to calculate the rate constants (*k*_1_) and half-lives (t_1/2_). The bacterial concentrations were as follows: free cells (2 × 10^7^ CFU ml^−1^), alginate-Ca immobilized particles (1.0 × 10^8^ CFU g^−1^), and alginate-Ca/WPU immobilized particles (1.2 × 10^8^ CFU g^−1^). According to the results shown in Table 5, the optimal temperature for DBP biodegradation was 30 °C. The optimal DBP biodegradation rate constant of *B. aquimaris* at 30 °C, calculated using first-order kinetics, was 0.18 day^−1^ with a half-life of 1.82 days when utilizing alginate-Ca/WPU immobilized particles. These findings indicated that *B. aquimaris* demonstrates optimal DBP biodegradation performance at 30 °C with alginate-Ca/WPU immobilized particles, showcasing the highest degradation efficiency.

Based on the results shown in Table 5, the optimal pH level for DBP biodegradation by *B. aquimaris* was found to be 7. The DBP biodegradation rate constant of *B. aquimaris* at pH 7, calculated using first-order kinetics, was 0.38 day^−1^ with a half-life of 1.81 days when utilizing alginate-Ca/WPU immobilized particles. These findings indicated that *B. aquimaris* demonstrates optimal DBP biodegradation performance under neutral pH conditions, particularly at pH 7 with alginate-Ca/WPU immobilized particles, showcasing the highest degradation efficiency.

In the context of adding various carbon and nitrogen sources, according to the results presented in Table 5, the optimal additional carbon and nitrogen sources for DBP biodegradation by *B. aquimaris* were yeast extract (0.15 g) and glucose (0.0625 g). The DBP biodegradation rate constant of *B. aquimaris* under these conditions, calculated using first-order kinetics, was 0.54 day^−1^ with a half-life of 1.28 days when utilizing alginate-Ca/WPU immobilized particles. These findings indicated that *B. aquimaris* demonstrates optimal DBP biodegradation performance when additional carbon and nitrogen sources are added to the culture medium with alginate-Ca/WPU immobilized particles, showcasing the highest degradation efficiency.

Based on the results presented in Table 5, the optimal surfactant for enhancing DBP biodegradation by *B. aquimaris* was Triton X-100 (1 CMC). Under these conditions, the DBP biodegradation rate constant of *B. aquimaris*, calculated using first-order kinetics, was 0.32 day^−1^ with a half-life of 2.16 days when utilizing alginate-Ca/WPU immobilized particles. These findings underscored the superior DBP biodegradation performance of *B. aquimaris* in the presence of Triton X-100 (1 CMC) in the culture medium with alginate-Ca/WPU immobilized particles, demonstrating the highest degradation efficiency.

According to the above results, the optimal cultural conditions for *B. aquimaris* in DBP biodegradation were as follows: incubation at 30 °C and pH 7, with the additional supplementation of 0.15 g of yeast extract, 0.0625 g of glucose, and 1 CMC of Triton X-100. Table 5 illustrated the results of DBP biodegradation conducted under these optimal cultural conditions in both the culture medium and original river water. The results demonstrate that alginate-Ca/WPU immobilized particles exhibit the best DBP biodegradation capability among free cells and the two types of immobilized particles, regardless of the medium used. Comparing the optimal half-life of DBP biodegradation among free cells and the two types of immobilized particles in culture medium and original river water reveals slightly lower degradation efficiency in original river water. This discrepancy can be attributed to the presence of nutrients and a native microbial community in the river water environment. However, the optimal half-life of DBP biodegradation for alginate-Ca/WPU immobilized particles in culture medium and original river water conditions is consistent, at 1.61 and 1.65 days, respectively, indicating robust performance across different environments. Therefore, this innovative microbial immobilization method using alginate-Ca/WPU effectively shields *B. aquimaris* from ecological competition with indigenous microorganisms, thereby maintaining DBP biodegradation capability akin to that observed in culture medium.

### 3.5. Algal Biotoxicity of DBP

The biotoxicity of DBP before and after biodegradation by the alginate-Ca/WPU immobilized particles was assessed using *C. vulgaris*. The median effective concentration (EC_50_) of DBP for *C. vulgaris* was determined to be 4.6 mg L^−1^. Figure 4 illustrates the percentage of chlorophyll *a* in *C. vulgaris* compared to the blank following DBP biodegradation. The blank test represents biotoxicity analysis conducted without DBP treatment. The results show an increase in chlorophyll *a* content in *C. vulgaris* after biodegradation, indicating a marked decrease in biotoxicity following the removal of DBP from the culture medium. Compared to the blank, the chlorophyll *a* content of *C. vulgaris* in the DBP solution after biodegradation under optimal conditions in original river water was 85.4%. These results indicate that DBP biotoxicity can be significantly reduced through biodegradation under optimal culture conditions.

### 3.6. Microbial Community Analysis

The NGS analysis results are depicted in Figure 5. In the NGS analysis of the free cells group, it is observed that from day 0 of adding *B. aquimaris* culture into the original river water, up to seven days later, *Bacillus* genus bacteria were not predominant in either NGS analysis result. Given that the free cells group involved direct extraction and analysis from the original river water, the detected *Bacillus* genus bacteria likely included indigenous *Bacillus* species present in the original river water in addition to the introduced *B. aquimaris*. Therefore, the NGS results indicated that the direct introduction of DBP-degrading strains into a contaminated environment is consistently subject to ecological competition from the native microbial community.

In the NGS analysis of the alginate-Ca and alginate-Ca/WPU groups, it is observed that, at day 0, when both sets of immobilized particles were introduced into the original river water, the internal NGS analysis of both groups showed *Bacillus* genus bacteria to be the predominant microbial community, with relative abundances (%) of 97.68% for the alginate-Ca group and 96.92% for the alginate-Ca/WPU group. However, in the NGS analysis results on day seven, *Bacillus* genus bacteria remained predominant in the alginate-Ca/WPU group with a relative abundance (%) of 87.33%, while in the alginate-Ca group, *Bacillus* genus bacteria were no longer predominant, with a reduced relative abundance (%) of 23.97%, replaced instead by *Pseudomonas* genus bacteria with a relative abundance (%) of 31.32%. These results indicated that the innovative microbial immobilization method employed in producing alginate-Ca/WPU immobilized particles restricts the infiltration of indigenous microorganisms from the external environment, thereby minimizing ecological competition and preserving the optimal DBP biodegradation capability.

## 4. Discussion

Biochar produced at 600 °C tends to exhibit alkaline properties than biochar produced at 300 °C. This phenomenon arises from the promotion of the decomposition of acidic functional groups, such as carboxyl and phenolic hydroxyl groups, and the volatilization of organic acids during pyrolysis at elevated temperatures when preparing biochar from herbs and woody plants. As a result, most agricultural and forestry wastes yield biochar with a pH above 7.0. Alkaline components like carbonates and hydroxides are preserved during pyrolysis, leading to an increase in biochar pH with higher pyrolysis temperatures [44]. Additionally, biochar, being alkaline, exerts significant effects in ameliorating acidic soil conditions. Consequently, biochar holds considerable promise for addressing contemporary challenges associated with agricultural soil acidification and base ion depletion.

The CEC of biochar is influenced by both pyrolysis temperature and raw materials used. High pyrolysis temperatures facilitate the oxidation of aromatic carbon and the formation of carboxyl functional groups [45]. These functional groups (-COOH, -OH) contribute to an increase in the negative charge on the biochar surface, thereby elevating the CEC. It has been observed that higher temperatures and anaerobic conditions yield a small amount of syngas (~20%) and a significant quantity of bio-oil (~60% of mass) and PAHs. Partial aerobic combustion of raw biomass during biochar production at high temperatures (pyrolysis) can also lead to PAH formation [46] and recondensation of several volatile organic compounds [47]. Production of biochar at higher temperatures might pose toxicity concerns for the microbial coating on the biochar’s studied in this research. Labile carbon (readily oxidizable carbon, ROC) content decreased with the increasing pyrolysis temperature of the biochar, as indicated by the reduction in the mass% of oxygen (Table 2). To preserve the activity of microbial particles on biochar, we opted for biochar produced at lower temperatures.

In previous studies, *B. aquimaris* has been demonstrated to degrade various pollutants, including microplastics [48,49]. This strain can be isolated from soil or marine environments and is commonly found in the environment [48,50]. Previous research has extensively used alginate-Ca [51,52] and WPU [53,54] in immobilized bacterial biodegradation. However, the innovative microbial immobilization method reported here, involving adsorption of bacteria into pores on biochar surfaces followed by sequential encapsulation with alginate-Ca and WPU has not been previously reported. Analysis from this study reveals that when *B. aquimaris*, possessing DBP degradation capability, is encapsulated within alginate-Ca/WPU immobilized particles, its dual-layered chemical structure restricts the infiltration of indigenous microorganisms from the external environment. However, DBP, with a molecular size much smaller than microorganisms, can still enter the particles and be degraded by B. aquimaris. In contrast, free cells, not encapsulated by any immobilization matrix, experience reduced DBP biodegradation capability due to ecological competition. This approach ensures that when applied in real contaminated environments, *B. aquimaris* maintains its DBP biodegradation capability without experiencing a decline due to ecological competition [55], thus preserving its efficacy similar to that observed in laboratory culture medium.

In this study, an inorganic microbial culture medium supplemented with DBP as the carbon source for *B. aquimaris* was employed. However, the addition of extra carbon and nitrogen sources could boost bacterial metabolism and growth rates. Therefore, glucose was used as the additional carbon source, along with yeast extract as the additional nitrogen source. The results indicate that the inclusion of yeast extract (0.15 g) and glucose (0.0625 g) in the culture medium significantly enhanced the DBP biodegradation rate of *B. aquimaris*, potentially optimizing the C/N ratio for this bacterium in the system. Additionally, as shown in Table 5, the use of triton X-100 resulted in increased DBP biodegradation by *B. aquimaris* compared to other surfactants. Triton X-100, a non-ionic surfactant with a molecular weight of 624, has a CMC of 0.23 mM. Under triton X-100 treatment, DBP partitioned into the micellar phase of triton X-100 became readily accessible to microbial action, leading to enhanced biodegradation.

In conclusion, the innovative microbial immobilization strategy utilizing biochar-alginate-Ca/WPU composites has proven to be highly effective in enhancing the biodegradation of DBP. The study identified *B. aquimaris* as a key DBP-degrading bacterium and optimized the cultivation conditions for maximum degradation efficiency. The composite materials provided an ideal environment for the growth and activity of the bacteria, resulting in a significant decrease in DBP concentrations. Moreover, algal biotoxicity analysis demonstrated a notable reduction in biotoxicity following DBP removal, indicating the potential ecological benefits of this remediation approach. NGS analysis revealed dynamic changes in the microbial community within the immobilized particles during the DBP degradation process, highlighting the effectiveness of the immobilization method. Overall, this study offers a novel and promising strategy for the rapid and efficient removal of DBP from contaminated environments, and also emphasizes the importance of microbial immobilization techniques in environmental remediation efforts.

## Figures and Tables

**Figure 1 microorganisms-12-01265-f001:**
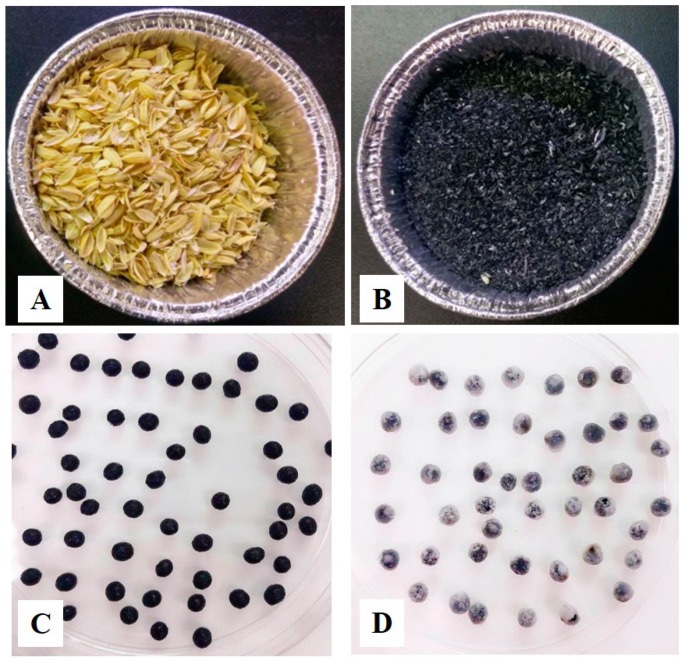
Appearance of RH, RHB, Alginate-Ca, and Alginate-Ca/WPU. (**A**) RH; (**B**) RHB; (**C**) Alginate-Ca; (**D**) Alginate-Ca/WPU.

**Figure 2 microorganisms-12-01265-f002:**
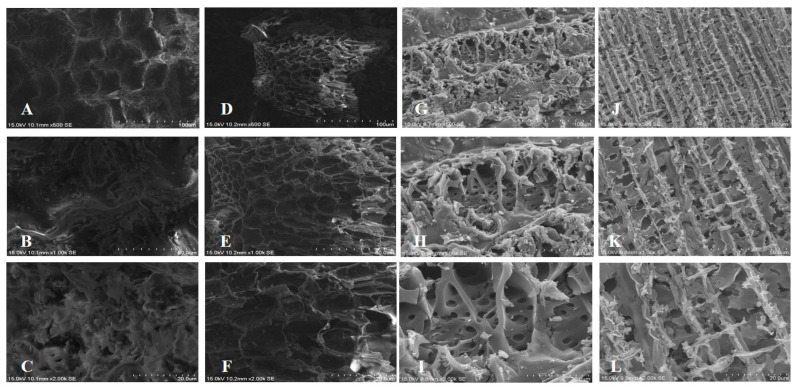
SEM images of RH and RHB produced at different pyrolysis temperatures. (**A**–**C**) show RH at magnifications of 500×, 1000×, and 2000×, respectively; (**D**–**F**) show RHB-300 at magnifications of 500×, 1000×, and 2000×, respectively; (**G**–**I**) show RHB-450 at magnifications of 500×, 1000×, and 2000×, respectively; and (**J**–**L**) show RHB-600 at magnifications of 500×, 1000×, and 2000×, respectively.

**Figure 3 microorganisms-12-01265-f003:**
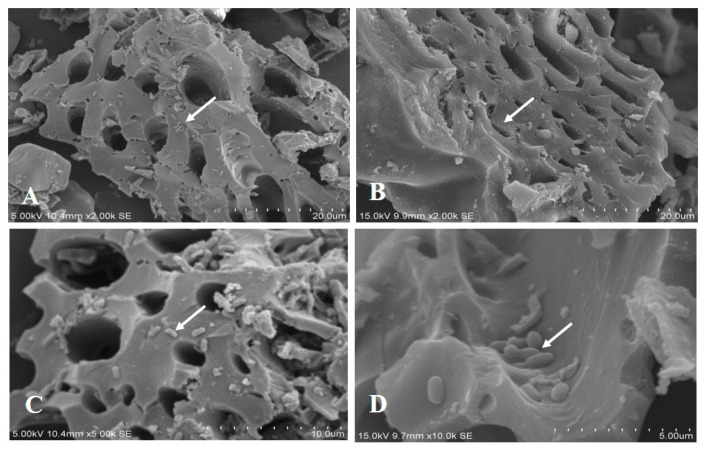
SEM images of RHB-300 which has immersed the bacterial suspension for 48 h. (**A**) 2000×; (**B**) 2000×; (**C**) 5000×; and (**D**) 10,000×.

**Figure 4 microorganisms-12-01265-f004:**
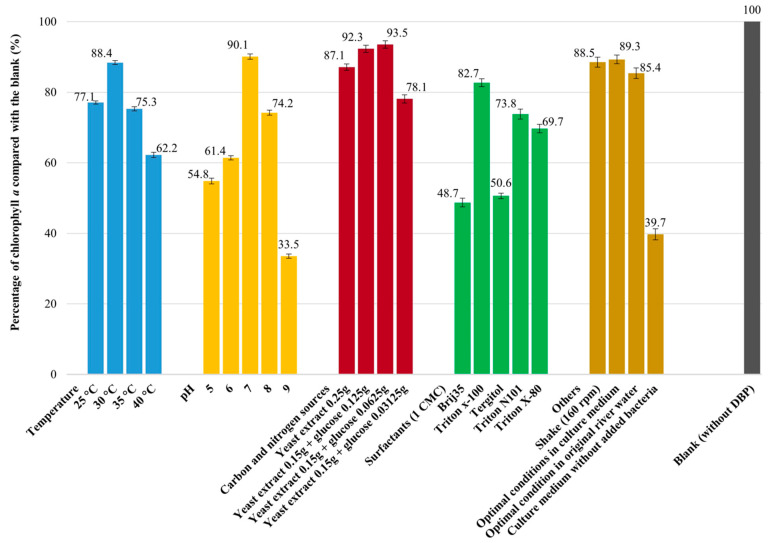
Percentage of chlorophyll *a* in *C. vulgaris* compared with the blank after DBP biodegradation by alginate-Ca/WPU immobilized particles. Data from three measurements are presented as mean ± SE.

**Figure 5 microorganisms-12-01265-f005:**
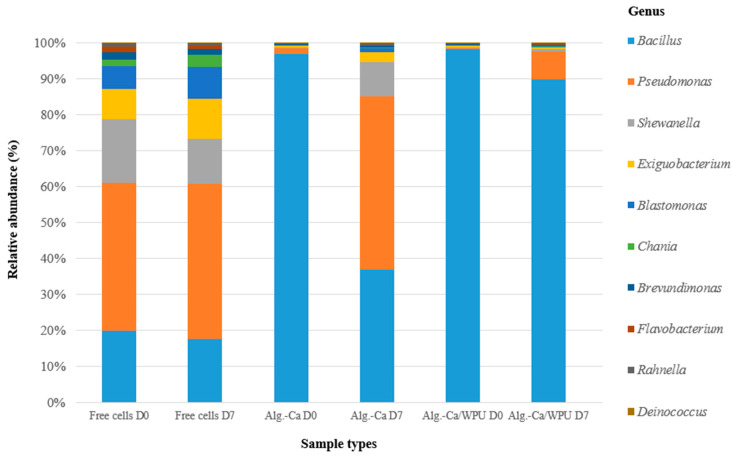
NGS analysis of free cells and two immobilized particles at the genus levels in original river water before and after 7 days of DBP biodegradation.

**Table 1 microorganisms-12-01265-t001:** Basic properties of the RH and RHB used in this study.

	RH	RHB-300	RHB-450	RHB-600
pH (1:10 *w*/*v*)	5.92	7.65	7.99	8.03
Total carbon (%)	40.5	31.2	28.2	26.8
Total nitrogen (%)	0.57	0.41	0.32	0.28
C/N ratio	71.1	76.1	88.1	95.7
Water content (%)	15.9	9.6	8.4	7.3
CEC (cmol_(+)_ kg^−1^)	46.1	24.2	26.2	35.5

ND: not determined; CEC: cation exchange capacity; Ca, Mg, K, and Na were determined as the exchangeable forms. RHB-300, RHB-450, and RHB-600 are the rice hull biochars produced at 300 °C, 450 °C, and 600 °C, respectively.

**Table 2 microorganisms-12-01265-t002:** Chemical compositions (mass%) of RH and RHB surfaces determined using EDS.

Element	RH		RHB-300		RHB-450		RHB-600	
	Weight	Atomic	Weight	Atomic	Weight	Atomic	Weight	Atomic
C	77.97	83.68	63.97	70.52	77.64	83.28	76.58	82.73
O	17.88	14.41	35.08	29.03	18.76	15.11	18.59	15.08
Si	4.08	1.87	0.95	0.45	3.27	1.50	4.57	2.11
K	0.08	0.03	0.00	0.00	0.24	0.08	0.18	0.06
Ca	0.00	0.00	0.00	0.00	0.09	0.03	0.07	0.02
Mn	0.00	0.01	0.00	0.00	0.00	0.00	0.00	0.00
Totals	100.00		100.00		100.00		100.00	

**Table 3 microorganisms-12-01265-t003:** Water content and weight of single microbial immobilized particle produced using different ratios of alginate-Ca.

Alginate-Ca (%)	Water Content (%)	Weight of Single Particle (mg)
0.5	18.96	31.31 ± 0.85
2	26.45	55.94 ± 1.34
3	35.66	68.65 ± 1.88

**Table 4 microorganisms-12-01265-t004:** Weight loss test result of microbial immobilized particles.

Alginate-Ca (%)	Initial Weight (g)	Weight after 48 h Shaking (g)	Weight Loss Ratio (%)	Weight after 96 h Shaking (g)	Weight Loss Ratio (%)
0.5	5	4.39 ± 0.22	12.23	3.85 ± 0.12	23.17
2	5	4.47 ± 0.21	10.52	4.08 ± 0.14	18.42
3	5	4.59 ± 0.22	8.23	4.24 ± 0.15	15.21

**Table 5 microorganisms-12-01265-t005:** Comparison of the effects of various incubation factors on DBP biodegradation rate constants (*k*_1_, day^−1^) and half-lives (t_1/2_, days) by free cells and two immobilized particles.

Treatment	Free Cells			Alginate-Ca			Alginate-Ca/WPU		
	*k*_1_ (day^−1^)	t_1/2_ (days)	r^2^	*k*_1_ (day^−1^)	t_1/2_ (days)	r^2^	*k*_1_ (day^−1^)	t_1/2_ (days)	r^2^
Temperature									
25 °C	0.16	4.42	0.91	0.19	3.57	0.88	0.28	2.39	0.85
30 °C	0.18	3.89	0.88	0.22	3.12	0.84	0.38	1.82	0.92
35 °C	0.14	5.11	0.88	0.16	4.31	0.87	0.27	2.61	0.77
40 °C	0.15	4.61	0.85	0.13	4.95	0.82	0.22	3.09	0.72
pH									
5	0.11	6.32	0.84	0.13	5.39	0.91	0.19	3.61	0.85
6	0.13	5.23	0.94	0.18	3.69	0.87	0.22	3.16	0.84
7	0.18	3.88	0.88	0.22	3.13	0.84	0.38	1.81	0.91
8	0.15	4.59	0.84	0.15	4.34	0.86	0.26	2.55	0.77
9	0.11	6.41	0.89	0.08	8.11	0.89	0.12	5.62	0.85
Carbon and nitrogen sources									
Yeast extract 0.25 g	0.31	2.29	0.89	0.45	1.54	0.87	0.38	1.81	0.86
Yeast extract 0.15 g + glucose 0.125 g	0.28	2.49	0.92	0.39	1.77	0.89	0.47	1.46	0.83
Yeast extract 0.15 g + glucose 0.0625 g	0.34	2.03	0.85	0.45	1.52	0.86	0.54	1.28	0.79
Yeast extract 0.15 g + glucose 0.03125 g	0.15	3.77	0.92	0.32	2.11	0.81	0.31	2.29	0.86
Surfactants (1 CMC)									
Brij 35	0.12	5.58	0.83	0.14	4.76	0.88	0.15	4.41	0.92
Triton X-100	0.23	2.99	0.95	0.25	2.81	0.94	0.32	2.16	0.93
Tergitol	0.16	4.22	0.81	0.20	3.42	0.75	0.16	4.30	0.87
Triton N101	0.16	4.46	0.86	0.16	4.19	0.85	0.27	2.59	0.78
Triton X-80	0.22	3.22	0.78	0.16	4.21	0.95	0.25	2.81	0.92
Others									
Shake (160 rpm)	0.21	3.59	0.85	0.23	3.08	0.89	0.39	1.78	0.89
Optimal conditions in culture medium	0.27	2.53	0.98	2.04	3.12	0.97	0.43	1.61	0.94
Optimal condition in original river water	0.13	5.21	0.88	0.18	3.79	0.86	0.45	1.65	0.93
Culture medium without added bacteria	0.01	84.83	0.81	0.01	61.81	0.79	0.01	50.93	0.81

## Data Availability

Data are contained within the article.

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
