# Peer review of "Innovative Microbial Immobilization Strategy for Di-n-Butyl Phthalate Biodegradation Using Biochar-Calcium Alginate-Waterborne Polyurethane Composites"

_microorganisms, 2024, doi:10.3390/microorganisms12071265_

Round 1
Reviewer 1 Report
Comments and Suggestions for Authors "The paper covers a relevant topic, related to the biodegradation of plasticizers for water treatment. In general, the manuscript is well-structured. Thus, I am suggesting minor revision.Please see below some suggestions for conducting the revision:
Section 2.10 Please explain the statistical analysis. This reviewer thinks that the information in this section should be placed in the Results section.
Line 315-318: The paragraph “This phenomenon suggested that rice hull transformed into rice hull biochar with a higher C/N ratio post-pyrolysis. The reason behind this transformation may have involved the formation of various compounds or functional groups, including Polycyclic Aromatic Hydrocarbons (PAHs), on the surface of biochar after pyrolysis” needs clarification. Pyrolysis is indeed a thermochemical process that always changes both the bulk and surface chemistry of the material being pyrolyzed. This process involves the decomposition of organic compounds, leading to a material with a higher carbon content and a higher C/N ratio. Since this is expected, the authors should focus their attention on the surface properties that affect the biochar adsorption performance, such as surface area, pore size and distribution, surface functional groups, and crystallinity (since crystallinity is an indicator of the presence of surface defects).
Line 329-331 states that “The decrease in oxygen proportion on the biochar surface may lead to an increase in the adsorption of chemicals such as PAHs”. However, it is not clear why less oxygen content leads to adsorption increase. Please elaborate on this idea. Consider adding a discussion on expected chemical compounds that have been either removed or reduced in the biochar and that are, expectedly, responsible for PAHs sorption.
Line 331-334. The authors mention that “To avoid concerns regarding excessive compound presence on the biochar surface during the fabrication of microbial immobilized particles, our study opted for the relatively safer and more stable RHB-300 as the raw material for preparing microbial immobilized particles”. The reader could be lost after reading this statement because it is not clear what does “safer and more stable RHB-300” refer to. Biochar is a very stable material. However, it is not clear why the authors mention biochar safety. Also, please consider measuring/characterizing the surface functional groups, for example, using FTIR. The corresponding results will be useful to discuss this section.
Table 5. Why the degradation observed with immobilized bacteria is different from that with free cells? Does it indicate that the immobilization matrix or procedure affects the degradation process? Please explain/expand the discussion on the findings reported in this table.
Line 498 “Biochar produced at higher temperatures tends to exhibit alkaline properties”. Please provide the temperature value that leads to biochar's alkaline properties. The temperature used in the work (300 °C) is not considered a high pyrolysis temperature.
In line 520-531, the authors mention that “Analysis from this study reveals that when B. aquimaris, possessing DBP degradation capability, is encapsulated within alginate-Ca/WPU immobilized particles, its dual-layered chemical structure restricts the infiltration of indigenous microorganisms from the external environment”. Could you please explain why DBPs can enter the dual-layered structure while indigenous microorganisms cannot? "
Author Response
Revision Notes:
We thank the reviewer for his/her careful reading and valuable comments on the manuscript. We have taken the comments on board to improve and clarify the manuscript. Please find below a detailed point-by-point response to all comments. All responses to the questions are also highlighted in the manuscript.
Comments of Reviewer #1:
- Section 2.10 Please explain the statistical analysis. This reviewer thinks that the information in this section should be placed in the Results section.
Response: We would like to thank the reviewer for providing this comment. Section 2.10 of the statistical analysis describes our observation that the microbial degradation of DBP fits well with first-order kinetics. Based on the characteristics of first-order kinetics, such as rate dependence on a single reactant, exponential decay, and constant half-life, we derived statistical formulas for calculating the rate constants (k1) and half-lives (t1/2). We appreciate the reviewer's comment and have incorporated the relevant information into the Results section (Section 3.4.). Thank you for providing this valuable comment.
- Line 315-318: The paragraph “This phenomenon suggested that rice hull transformed into rice hull biochar with a higher C/N ratio post-pyrolysis. The reason behind this transformation may have involved the formation of various compounds or functional groups, including Polycyclic Aromatic Hydrocarbons (PAHs), on the surface of biochar after pyrolysis” needs clarification. Pyrolysis is indeed a thermochemical process that always changes both the bulk and surface chemistry of the material being pyrolyzed. This process involves the decomposition of organic compounds, leading to a material with a higher carbon content and a higher C/N ratio. Since this is expected, the authors should focus their attention on the surface properties that affect the biochar adsorption performance, such as surface area, pore size and distribution, surface functional groups, and crystallinity (since crystallinity is an indicator of the presence of surface defects).
Response: We would like to thank the reviewer for providing this comment. We have cited three previous articles and indicated some surface properties of rice hull biochar pyrolyzed with different temperatures. The revised sentences were made in Section 3.2. Thank you for providing this valuable comment.
Jien, S.H.; Wang, C.S. Effects of biochar on soil properties and erosion potential in a highly weathered soil. Catena. 2013, 110, 225-233.
Jien, S.H. Physical characteristics of biochars and their effects on soil physical properties. Biochar from biomass and waste. Elsevier, 2019, 21-35.
Lua, A.C.; Yang, T.; Guo, J. Effects of pyrolysis conditions on the properties of activated carbons prepared from pistachio-nut shells. J. Anal. Appl. Pyrolysis. 2004, 72, 279-287.
- Line 329-331 states that “The decrease in oxygen proportion on the biochar surface may lead to an increase in the adsorption of chemicals such as PAHs”. However, it is not clear why less oxygen content leads to adsorption increase. Please elaborate on this idea. Consider adding a discussion on expected chemical compounds that have been either removed or reduced in the biochar and that are, expectedly, responsible for PAHs sorption.
Response: We would like to thank the reviewer for providing this comment. We have revised the sentence as “As the pyrolysis temperature increased, the proportion of oxygen gradually decreased and specific surface area (SSA) increased. The increase in SSA decrease in oxygen pro-portion on the biochar surface may lead to an increase in the adsorption of chemicals such as PAHs.” In Section 3.2. Thank you for providing this valuable comment.
- Line 331-334. The authors mention that “To avoid concerns regarding excessive compound presence on the biochar surface during the fabrication of microbial immobilized particles, our study opted for the relatively safer and more stable RHB-300 as the raw material for preparing microbial immobilized particles”. The reader could be lost after reading this statement because it is not clear what does “safer and more stable RHB-300” refer to. Biochar is a very stable material. However, it is not clear why the authors mention biochar safety. Also, please consider measuring/characterizing the surface functional groups, for example, using FTIR. The corresponding results will be useful to discuss this section.
Response: We would like to thank the reviewer for providing this comment. We have revised the sentences in Line 341-342. Thank you for providing this valuable comment.
- Table 5. Why the degradation observed with immobilized bacteria is different from that with free cells? Does it indicate that the immobilization matrix or procedure affects the degradation process? Please explain/expand the discussion on the findings reported in this table.
Response: We would like to thank the reviewer for providing this comment. As stated in the last paragraph of Section 3.4, the difference in DBP biodegradation capability between immobilized bacteria and free cells is due to the "microbial immobilization method using alginate-Ca/WPU, which effectively shields B. aquimaris from ecological competition with indigenous microorganisms, thereby maintaining DBP biodegradation capability akin to that observed in culture medium." Additionally, the Discussion section explains that "analysis from this study reveals that when B. aquimaris, possessing DBP degradation capability, is encapsulated within alginate-Ca/WPU immobilized particles, its dual-layered chemical structure restricts the infiltration of indigenous microorganisms from the external environment." This occurs because the cross-linked structure of alginate-Ca and WPU blocks indigenous microorganisms from entering the immobilized particles and contacting B. aquimaris. However, DBP, with a molecular size much smaller than microorganisms, can still enter the particles and be degraded by B. aquimaris. In contrast, free cells, not encapsulated by any immobilization matrix, experience reduced DBP biodegradation capability due to ecological competition. The reviewer's comment is appreciated, and the relevant information has been incorporated into the Discussion section (line 543-546). Thank you for providing this valuable comment.
- Line 498 “Biochar produced at higher temperatures tends to exhibit alkaline properties”. Please provide the temperature value that leads to biochar's alkaline properties. The temperature used in the work (300 °C) is not considered a high pyrolysis temperature.
Response: We would like to thank the reviewer for providing this comment. We have provided the temperature and revised the sentences in Line 509-510. Thank you for providing this valuable comment.
- In line 520-531, the authors mention that “Analysis from this study reveals that when B. aquimaris, possessing DBP degradation capability, is encapsulated within alginate-Ca/WPU immobilized particles, its dual-layered chemical structure restricts the infiltration of indigenous microorganisms from the external environment”. Could you please explain why DBPs can enter the dual-layered structure while indigenous microorganisms cannot? "
Response: We would like to thank the reviewer for providing this comment. As mentioned in the response to Question 5, indigenous microorganisms cannot enter the immobilized particles due to the cross-linked structure of alginate-Ca and WPU, which acts as a barrier. However, DBP can enter because it is a compound with a molecular size much smaller than microorganisms. In preliminary experiments of this study, it was tested whether DBP entering the particles was being adsorbed by alginate-Ca or WPU or degraded by B. aquimaris inside the particles. The presence of phthalic acid, a metabolic product of DBP, detected using LC/MSMS, confirmed that the decrease in DBP concentration in the culture system was indeed due to biodegradation. Thank you for providing this valuable comment.
We thank the reviewer for the constructive and insightful comments, which have helped us to substantially improve our manuscript.

Reviewer 2 Report
Comments and Suggestions for Authors
The manuscript entitled “Innovative Microbial Immobilization Strategy for Di-n-Butyl Phthalate Biodegradation Using Biochar-Calcium Alginate-Waterborne Polyurethane Composites» is devoted to an important problem Di-n-butyl phthalate (DBP) biodegradation which is a prevalent phthalate ester widely used as a plasticizer. In this study authors used Bacillus aquimaris strain immobilized on rice gbiochar, calcium alginate and waterborne polyurethane (WPU) composites to enhance the biodegradation efficiency of DBP. To my mind this manuscript is topical and corresponding to the aims and scopes of the “Microorganisms” journal.
General and specific Comments.
1. The abstract should be more specific and show the results obtained by the authors. It should not use phrases like “innovative microbial immobilization strategy, This novel approach etc”; it does not serve the purpose of advertising your products, but shows the scientific community the results of the research presented in the manuscript.
2. Once you make a diagram for immobilizing bacteria, it is difficult to understand from the text. Kind of like what you did in graphical abstract.
3. In section 3.1. it is said that Twenty-five bacterial strains are capable of utilizing DBP as the carbon source and energy were isolated from water and sediment samples, but no data is provided on their description and the efficiency of DBP consumption. What degradation efficiency values allowed you to select one strain?
4. Figure 1. Worth removing in saplimentary
5. 471 adding B. aquimaris culture into the original river 471 water? What do you mean? in situ test? This needs to be written down in detail in the methods. Were these immobilized cells? What is the chemical composition of river water? How much pollutant is added?
6. 511 on the biochar was the negative charge value measured?
7. In the discussion, it is worth giving a diagram of a grain particle and thinking about how accessible DBP is to the bacteria living in it, which of the components of the particle plays a more important role in its biological activity.
8. It is worth highlighting the conclusion section separately.
In general, the text lacks specific consumption values, comparing these values with other studies
Comments on the Quality of English LanguageMinor editing of English language required
Author Response
Revision Notes:
We thank the reviewer for his/her careful reading and valuable comments on the manuscript. We have taken the comments on board to improve and clarify the manuscript. Please find below a detailed point-by-point response to all comments. All responses to the questions are also highlighted in the manuscript.
Comments of Reviewer #2:
- The abstract should be more specific and show the results obtained by the authors. It should not use phrases like “innovative microbial immobilization strategy, This novel approach etc”; it does not serve the purpose of advertising your products, but shows the scientific community the results of the research presented in the manuscript.
Response: We would like to thank the reviewer for providing this comment. Based on the reviewer's suggestion, we have carefully revised the abstract. Thank you for providing this valuable comment.
- Once you make a diagram for immobilizing bacteria, it is difficult to understand from the text. Kind of like what you did in graphical abstract.
Response: We would like to thank the reviewer for providing this comment. Based on the reviewer's suggestion, we have added more detailed descriptions for the graphical abstract in the manuscript. In the future, we will design and create more illustrative graphical abstracts with greater care. Thank you for providing this valuable comment.
- In section 3.1. it is said that Twenty-five bacterial strains are capable of utilizing DBP as the carbon source and energy were isolated from water and sediment samples, but no data is provided on their description and the efficiency of DBP consumption. What degradation efficiency values allowed you to select one strain?
Response: We would like to thank the reviewer for providing this comment. In this section, we initially followed the incubation conditions mentioned in the manuscript: 30°C, pH 7.5, with an initial DBP concentration of 5 mg L⁻¹. Under these conditions, twenty-five bacterial strains were able to survive. Subsequently, using the analytical methods described in Section 2.7, the residual DBP in the culture medium of these 25 bacterial strains was analyzed. The results revealed that only three strains exhibited significant DBP degradation capability within 48 hours. Among these three, B. aquimaris (HJ-6) was identified as having the most significant DBP degradation capability. This information has been added to Section 3.1. Thank you for providing this valuable comment.
- Figure 1. Worth removing in saplimentary.
Response: We would like to thank the reviewer for providing this comment. After thorough discussion and evaluation by all authors, we collectively decided to retain Figure 1 in the manuscript. We kindly request the reviewer's understanding and approval to keep Figure 1 in the main manuscript rather than moving it to the supplementary section. Thank you for providing this valuable comment.
- 471 adding B. aquimaris culture into the original river 471 water? What do you mean? in situ test? This needs to be written down in detail in the methods. Were these immobilized cells? What is the chemical composition of river water? How much pollutant is added?
Response: We would like to thank the reviewer for providing this comment. We have been corrected this error and add the following sentences into the section 2.9. “After 48 hours of incubation, 5 mL of culture medium or 5 g of immobilized particles were added to 45 mL of original river water supplemented with 5 mg L⁻¹ DBP. DBP was not detectable or was below the detection limit in the original river water. The mixture was then incubated for 7 days to evaluate the survival of DBP-degrading bacteria in the natural environment”. Thank you for providing this valuable comment.
- 511 on the biochar was the negative charge value measured?
Response: We would like to thank the reviewer for providing this comment. We did not measured the surface charge of biochar in this study, however, we cited the reference [46] “Glaser, B.; Lehmann, J.; Zech, W. Ameliorating physical and chemical properties of highly weathered soils in the tropics with charcoal - a review. Biol. Fertility Soils. 2002, 35, 219–230” in this study. Thank you for providing this valuable comment.
- In the discussion, it is worth giving a diagram of a grain particle and thinking about how accessible DBP is to the bacteria living in it, which of the components of the particle plays a more important role in its biological activity.
Response: We would like to thank the reviewer for providing this comment. This is an interesting and meaningful suggestion. We are currently drafting another article on rice husk biochar, and we will consider including such a comparison table in the next article. Thank you for providing this valuable comment.
- It is worth highlighting the conclusion section separately.
Response: We would like to thank the reviewer for providing this comment. We also fully agree with the reviewer's suggestion. However, the manuscript format for this journal requires the conclusion to be included within the Discussion section. We also greatly desire to have the conclusion as a separate section. Meanwhile, we have slightly revised the conclusion to make its content more specific. Thank you for providing this valuable comment.
We thank the reviewer for the constructive and insightful comments, which have helped us to substantially improve our manuscript.

Round 2
Reviewer 2 Report
Comments and Suggestions for Authors
the authors have significantly improved the manuscript, I am ready to recommend it for publication in this form